# Effects of Black Liquor Shocks on the Stability of Activated Sludge Treatment of Kraft Pulp Mill Effluent: Morphological Alteration in *Daphnia magna* and Mutagenicity and Genotoxicity Response in *Salmonella typhimurium*

**Soledad Chamorro** [1], **Laura Hernández** [1], **Katia Saéz** [2], **Gloria Gómez** [1] **and Gladys Vidal** [1,*]

[1] Center Europe Latin America (EULA)-Chile, Environmental Engineering & Biotechnology Group (GIBA-UDEC), Environmental Science Faculty, Universidad de Concepción, Concepcion 4070386, Chile; schamorr@gmail.com (S.C.); laura.hdez.gonzalez@gmail.com (L.H.); gloriagomezosorio@gmail.com (G.G.)

[2] Department of Statistics, Faculty of Mathematics and Physical Sciences, Universidad de Concepción, Concepcion 4070386, Chile; ksaez@udec.cl

[*] Correspondence: glvidal@udec.cl

**Abstract:** The objective of this study is to evaluate the stability of activated sludge (AS) in the treatment of kraft pulp mill effluent exposed to black liquor shock, as well as the effect of its exposure on the morphology of *Daphnia magna* and DNA damage through mutagenicity and genotoxicity response in *Salmonella typhimurium*. To this end, we applied doses of 2-, 4-, 10-, and 30-mL black liquor/L influent—shock 1 (S1), shock 2 (S2), shock 3 (S3), and shock 4 (S4), respectively—to kraft pulp mill effluent. The system stability was checked by analyzing heterotrophic biomass activity and discharge quality, evaluated using Ames test and *Daphnia magna*. The results show that the chemical oxygen demand (COD) removal efficiency for normal conditions was 64.84%, falling to 61.68%, and 61.31% for S1 and S2, respectively, and values of 52.11% for S3 and 20.34% for S4. The biomass activity decreased after each shock was applied, but then recovered. There was no evidence of lethal toxicity ($LC_{50}$) to *Daphnia magna* at any of the concentrations. Therefore, it is feasible to apply doses S1 and S2 to an AS system that treats kraft pulp mill effluent.

**Keywords:** black liquor toxicity; activated sludge performance; organic matter biodegradation; heterotrophic biomass activity; effluent toxicity

## 1. Introduction

The treatment of kraft pulp mill effluent is essential for reducing pollutant load and complying with environmental legislation and standards. Conventional treatment technologies such as activated sludge (AS) and moving-bed biofilm reactors (MBBR) are currently installed in kraft pulp mills to remove organic compounds and acute toxicity from the kraft mill effluents [1,2]. The AS aerobic degradation of organic compounds ranges between 50 and 65% of the chemical oxygen demand (COD) [1,3].

Despite the existence of these technologies, effluent discharges from kraft pulp mills have been identified as potential contaminants of aquatic environments [4]. Their toxic nature results from the presence of several naturally occurring and xenobiotic compounds, which are formed and released during various stages of the process [5,6]. For example, black liquor is generated in the digestion stage and characterized by inorganic cooking elements and degraded and dissolved wood substances [7,8]. The organic component consists of alkali lignin and the sodium salts of the polysaccharinic acids, phytosterols, resin acids, and fatty acids, which generate high levels of biological oxygen demand and chemical oxygen demand [9]. In the process, the chemicals used in digestion are regenerated in a closed cycle, reducing environmental impacts by diverting the black liquor from the pulp mill effluent treatment system [2]. Black liquor spills are accidental and are retained in spill

systems. However, after an accident, the only way to discharge these spills is through an effluent treatment plant [2]. Many of the effects of effluents discharged by kraft mills are caused by unplanned discharges of black liquor, usually in concentrations that are low in relation to the values that are normal in the pulp mill or recovery areas [10]. Kraft mill effluents can contain compounds capable of affecting endocrine systems. Despite the primary and secondary effluent treatment systems implemented by most mills, a series of studies pointed to wood-derived intermediate products that could be formed and released during normal mill operations, including chlorocimenes, resin acids, chloroethens, flavonoids, and phytoestrogens. Some of these compounds were classified as endocrine-disrupting compounds (EDCs) and were suspected to accumulate in sediment, potentially becoming bioavailable to fish, for example [1,11]. Direct and indirect toxicity assessments, making use of ecotoxicological tests, can play an important role in supporting decision making [12]. The fate and effects of kraft pulp mill effluent have been evaluated using diverse organisms, including *Daphnia* sp. [13]. This genus is widely used as a model organism in aquatic ecology, evolutionary biology, and ecotoxicology. *Daphnia* sp. are used as model organisms because they play a key ecological role in the food web dynamics of the pelagic zone. They are an important food resource for many fish species and are the main grazers on phytoplankton assemblages, thereby assuming a pivotal role in trophic interactions in freshwater ecosystems [14]. The classical bioassays using this species include acute toxicity assays (mortality 24–48 h). However, assays that focus on morphological alterations are valid for analyzing as sublethal responses, which are particularly important in functional animal biology and ecology [15,16]. *Daphnia* sp. present a multitude of inducible morphological shapes in response to changing regimes [17]. Reported changes in life-history are likely to be improved via adjustments in energy assimilation strategies by *Daphnia* sp. Studies that explain the results from the balance of investments in re-production, growth, and/or maintenance of *Daphnia* sp. have been made by Hart and Bychek [15]; meanwhile, studies regarding the change in size and shape of the head and body in *Daphnia* sp. were carried out by Preisser and Orrock [18,19] and Otte et al. [20]. Several types of mutagenic compounds have been identified in the pulp kraft mill effluent [20]. High-molecular-weight compounds generate organic matter that is difficult to biodegrade in biological systems; therefore, color is present in the final effluent [1,21,22]. However, the low-molecular-weight compounds, below 1000 g/mol, pose the greatest toxicological, mutagenic, and carcinogenic risk, as they easily penetrate the cell membrane [23–25]. Therefore, by damaging the DNA of organisms, genotoxins can initiate a cascade of impairments at the molecular, cellular, organ, whole organism, or population and community levels. DNA and cytogenetic alterations in aquatic organisms have been associated with an impaired enzyme function or general metabolism, cytotoxicity, immunotoxicity, abnormal development, and reduced survival, growth, and reproduction potency [23]. The objective of this study is to evaluate the stability of AS in the treatment of kraft pulp mill effluent exposed to black liquor shock, as well as the effect of its exposure to the morphology of *Daphnia magna* and DNA damage through mutagenicity and genotoxicity responses in *Salmonella typhimurium*.

This study is novel because a problem that arises in kraft pulp mills could be solved with the infrastructure installed (through the installed treatment plant), by adjusting an optimal process for dosing the black liquor spill to the activated sludge system. In this way, maintaining a stable operation in the activated sludge system will generate discharges suitable for surface ecosystems. In addition to evaluating the stability of the AS operation with biomass activity tools, this study also provides results on the toxicity of the treated effluent (mixture of the effluent from the plant process plus a dose of the black liquor spill) discharged into ecosystems.

## 2. Materials and Methods

### 2.1. Influent

The influent was obtained from a kraft pulp mill that processes *Eucalyptus globulus* and *Eucalyptus nitens* with an elemental chlorine-free (ECF) bleaching system. The samples

were obtained after primary wastewater treatment, which consisted of a settling tank to reduce fiber and suspended solids. The influent was stored in 30 L PVC tanks, which were refrigerated in a dark room at 4 °C.

## 2.2. Activated Sludge System

A lab-scale activated sludge (AS) system was implemented. It included an aerobic reactor (1.20 L) and a glass settling unit (0.39 L). To achieve a $BOD_5$:N:P proportion of 100:5:1 in the influent, nitrogen was supplemented with urea, but it was not necessary to add phosphorus [3]. The pH was adjusted to approximately 7 using HCl (0.5 N). The AS system was operated at a temperature of $22.01 \pm 2.01$ °C. The dissolved oxygen (DO) concentration was maintained at values higher than 2 mg/L using a diffuser aeration system [3]. The system recirculation was kept at 1.4 times the inlet flow rate.

The operation was divided into two phases: normal operation (C), fed with crude influent for two weeks, and the toxic shock phase, with black liquor added to the influent to destabilize the system. Black liquor from a local kraft pulp mill was added to the feed of the system in four trials, over 24 h. The trials were S1 (shock 1), S2 (shock 2), S3 (shock 3), and S4 (shock 4), and the black liquor concentrations added were 2, 4, 10, and 30 mL of black liquor/L of influent, respectively. After 24 h of each trial, the system was fed with influent without black liquor.

The AS system was operated continuously for 44 days. The operation included 14 days for start-up with an HRT of 2 days. During the next 30 days, HRT was kept at 1 day. Throughout the operational period, the efficiencies of S1–S4 were estimated for COD, biological oxygen demand within 5 days ($BOD_5$), total phenolic compounds, color, lignin and derivatives, lignosulfonic acids, and aromatic compounds, in accordance with:

$$E\ (\%) = \frac{Q_i \times C_i - Q_o \times C_o}{Q_i \times C_i \times 100} \tag{1}$$

where $E$ (%) is the removal percentage, $Q$ the flow rate (L/d), and $C$ the parameter concentration (mg/L); subindices "$i$" and "$o$" are the inflow and outflow, respectively.

## 2.3. Heterotrophic Biomass Activity

To evaluate biomass performance under normal and toxic conditions, heterotrophic biomass activity was measured weekly at 0 h, 2 h, 6 h, 24 h, and 48 h during the trials. Biomass activity was analyzed using the oxygen uptake rate (OUR) and the specific oxygen uptake rate (SOUR) according to Morales et al. [3].

## 2.4. Analytical Methods

The physicochemical parameters of COD (colorimetric method, 5210-B), $BOD_5$ (modified Winker azide method, 5210-B) were measured following standard methods [26]. The total phenolic compound (UV phenol) concentration was measured by UV absorbance in a $1 \times 1$ cm quartz cell at 215 nm and pH 8.0 (0.2 M $KH_2PO_4$ buffer) and transformed into a concentration using a calibration curve with phenol as the standard solution. Samples were membrane-filtered (0.45 μm). Color was measured at wavelengths of 440 nm, lignosulfonic acid at 346 nm, aromatic compounds at 254 nm, and derived lignin at 280 nm in a $1 \times 1$ cm quartz cell using a model Spectronic Unicam UV-Visible Series GenesysTM 10, in accordance with Chamorro et al. [27].

## 2.5. Mutagenicity Test

The Muta-ChromoPlate kit (EBPI Labs) 'Ames Test' [26] employs a mutant strain of *Salmonella typhimurium*. The strain *S. typhimurium* TA100 with metabolic activation (S9 liver extract of rats treated with Aroclor 1260) was used. Bacteria were grown in nutrient broth shaken overnight at 37 °C. The positive control was 2-aminofluorene for strains with S9 (20 μg/plate), and DMSO was tested as a negative control. Measures of 2.5 mL of the "reaction mixture" and 5 μL of tester bacterial cultures were mixed. After shaking, the

materials were transferred into wells on a 96-well plate (200 μL/well). The plates were incubated in the dark for 5 days at 37 °C. The surviving colonies and revertants were then counted. EBPI provides a template for statistical analysis. The S factor was calculated as the number of revertants per treatment over the number of revertants per positive control (+control).

### 2.6. Toxicity Testing in Daphnia magna

Female *D. magna* specimens were obtained from the Bioassay Laboratory, Environmental Sciences Faculty at the University of Concepcion. Stock cultures and bioassays were maintained at 20 ± 2 °C with a photoperiod of 16 h light and 8 h dark. The daphnids were fed three times per week and the chamber water was changed every 48 h, which favors parthenogenetic population growth. The feeding regimen and reconstituted water followed USEPA guidelines [28]. Acute toxicity tests (24–48 h) were conducted with S1, S2, S3, and S4 (<24 h old) following standard procedures for toxicity bioassays [28]. Each bioassay was performed in quadruplicate and exposed five individuals per bottle to 30 mL of sample solution in 50 mL polypropylene cups without food. Five dilutions were evaluated per sample, S1–S4 (6.25%, 12.5%, 25%, 50%, and 100%). At the end of each exposure, the lethal concentration ($LC_{50}$) was determined as the concentration that generated mortality or non-motility in 50% of the *D. magna* population. The significance of differences in the reproductive parameter was determined by Probit statistical program [28].

### 2.7. Morphological Alterations in Daphnia magna

*Daphnia magna* were observed using a light microscope fitted with a photographic camera. The daphnids were placed on a glass microscope slide, immobilized by removing the medium from the slide, and anatomical development was recorded in photographs. The dimensions of abdominal cavities (measured as the largest length of the lateral immobilized organism) were measured under 2× magnification. Total body length, defined as the distance from the top of a head capsule to the base of the shell spine, was also measured at the same time. The daphnids were photographed to observe the modification endpoints, such as the rostrum and caudal spine. Four replicates (five organisms) were performed for each treatment (100%, 50%, 25%, 12.5%, and 6.25%) in S1, S2, S3, and S4, consisting of 2 mL, 4 mL, 10 mL, and 30 mL of black liquor/L of influent, respectively, in addition to the control. Similarly, *Daphnia magna* body and abdomen length were measured at end of the bioassay (48 h). Body length was determined as the distance from the top of the eye to the base of the caudal spine [18].

### 2.8. Statistical Analysis

Bioassay statistical analyses were performed according to USEPA guidelines for lethal toxicity testing [28]. Each test was assessed for media lethal concentration ($LC_{50}$). Black liquor shock concentrations, dilution of lethal bioassay, and caudal spine and rostrum alteration were analyzed using a binomial logistic regression. The critical *p*-value for all experiments was 0.001. *Daphnia magna* body and abdomen length, analyzed by multiple regression, were used to determine the relationships between variables (black liquor shock concentration, bioassay dilution, and *Daphnia magna* body and abdomen length). The statistical treatment of the data was performed using "R" software.

## 3. Results and Discussion

### 3.1. Influent Characterization

Table 1 shows the physicochemical characteristics of bleached kraft pulp mill influent. This influent was used as food in normal conditions and was added to the different black liquor shocks (toxic shock conditions). The pH was maintained within the range of 7.10 ± 3.91, adequate for the survival and development of most aquatic organisms [29]. COD and $BOD_5$ presented values of 477.13 ± 76.10 mg/L and 345.0 ± 41.2 mg/L, respectively. The presence of high-molecular-weight compounds, such as lignin derivatives,

was observed, with average values of 2.12 ± 0.11 1 × 1 cm. Lignosulfonic acid presented values of 0.05 ± 0.02 1 × 1 cm, while color measured as absorbance at 440 mm presented a value of 0.1 ± 0.03 1 × 1 cm. Furthermore, the total phenolic compounds amounted to 150.30 ± 57.83 mg/L, which was consistent with the value found in Morales et al. [3]. Nitrogen and phosphorus concentrations were low in relation to those needed to maintain an adequate $BOD_5$:N:P balance to maintain the metabolism of aerobic bacteria in the degradation process of BOD5 in the AS. Thus, it is necessary to add nutrients to obtain the favorable degradation of kraft mill effluents.

**Table 1.** Physicochemical characteristic of bleached kraft mill influent.

| Parameter | Unit | Average ± SD |
|---|---|---|
| pH | - | 7.10 ± 3.91 |
| Conductivity | mS/cm | 3.05 ± 0.09 |
| $BOD_5$ | mg/L | 345.00 ± 41.20 |
| COD | mg/L | 477.13 ± 76.10 |
| Color (440 nm) | 1 × 1 cm | 0.15 ± 0.03 |
| Lignosulfnic acid (346 nm) | 1 × 1 cm | 0.05 ± 0.02 |
| Aromatic compounds (254 nm) | 1 × 1 cm | 0.59 ± 0.05 |
| Lignin derives (280 nm) | 1 × 1 cm | 2.12 ± 0.11 |
| Total phenolic compounds | mg/L | 150.30 ± 57.83 |
| TN | mg/L | <0.50 |
| TP | mg/L | 23.92 ± 0.59 |

$BOD_5$: biological oxygen demand within 5 days; COD: chemical oxygen demand; TN: total nitrogen; TP: total phosphorous. SD: standard deviation.

### 3.2. Black Liquor Trials

Table 2 presents the perturbation conditions for the activated sludge reactor. The black liquor is highly alkaline (pH greater than 12), while the pH was always maintained near to neutrality (7.50–7.12). Meanwhile, EC increased in direct relation with the concentration of black liquor fed into the reactor, from an initial condition (S1) of 3.47 mS/cm to 6.35 mS/cm in S4. A detailed study of a mill is required to assess the significance of unplanned liquor discharges, but the COD of the effluent provides a good first approximation for kraft mills [30]. In our study, the COD concentration increased from 708.00 ± 3.91 mg/L to 3221.00 ± 112.03 mg/L between S1 and S4, respectively. Considering that a value of 477.13 ± 76.10 mg/L for normal operations was obtained, increases of 148% (S1), 233% (S2), 395% (S3), and 675% (S4), respectively, were observed. Vadodaria [30] observed the effect of black liquor on the activated sludge process, reporting that black liquor affected all measured parameters of the mill effluent (conductivity, COD, TOC and pH).

**Table 2.** Characterization of kraft black liquor shocks.

| Shock | Black Liquor Dose (mL/L) * | Operation Time (d) | pH | EC (mS/cm) | COD (mg/L) | OLR Increase (%) |
|---|---|---|---|---|---|---|
| S1 | 2 | 13 | 7.5 | 3.47 | 708 ± 3.91 | 180.90 |
| S2 | 4 | 21 | 7.2 | 3.79 | 1113 ± 7.01 | 287.46 |
| S3 | 10 | 27 | 7.3 | 4.53 | 1887 ± 15.29 | 487.36 |
| S4 | 30 | 34 | 7.1 | 6.35 | 3221 ± 12.03 | 832.04 |

* mL of black liquor/L of influent; S1: shock 1; S2: shock 2; S3: shock 3; S4: shock 4; EC: electrical conductivity, COD: chemical oxygen demand, OLR: organic load rate.

### 3.3. Activated Sludge Performance

Figure 1 shows reactor behavior with respect to the removal efficiency of $BOD_5$, COD, color, and total phenolic compounds from the effluent. The hydraulic retention time (HRT) in the system was maintained for 24 h, corresponding to an OLR of around 0.34 ± 0.04 gCOD/L·d in normal conditions. For the toxic conditions, HRT was reduced stepwise from 24 h to 12 h, resulting in an increase in OLR up to 0.71 until 3.22 gCOD/L·d

was maintained with the application of each shock. The COD removal was 56.6% (average values) under the operation of AS without a shock of black liquor and with a peak of 64.84% COD removal. $BOD_5$ removal efficiency was 78.3 ± 6.5%, with a maximum of 85.1%. For toxic conditions, S1 presented COD removal of 61.68% and S2 and S3 presented average values of 61.31% and 52.11%, respectively. S4 presented the lowest efficiency (average value of 51.83%), with values as low as 20.34%. Clearly, in this condition, the AS system showed a destabilization of the system, due to this the decrease in the COD removal. The average removal efficiency of $BOD_5$ was up to 75% in all the shocks. This value is considered typical for activated sludge systems exposed to black liquor [30]. In general, these results are similar to those obtained by Xavier et al. [31], who observed the COD elimination of around 57.67% in activated sludge. Meanwhile, Diez et al. [32] showed that these processes eliminate a high percentage of $BOD_5$, between 85% and 90%. Nevertheless, COD elimination does not surpass 60%, due chiefly to the presence of high-molecular-weight lignin and phenolic chlorate compounds that are difficult to degrade [1,5]. Color removal efficiency was low, at less than 50.65%. The biological treatment method was not very effective at reducing color but proved satisfactory in $BOD_5$ and COD reduction. Finally, the average removal efficiency of total phenols was 24% in normal conditions (without the application of black liquor) and 28% when S4 was applied.

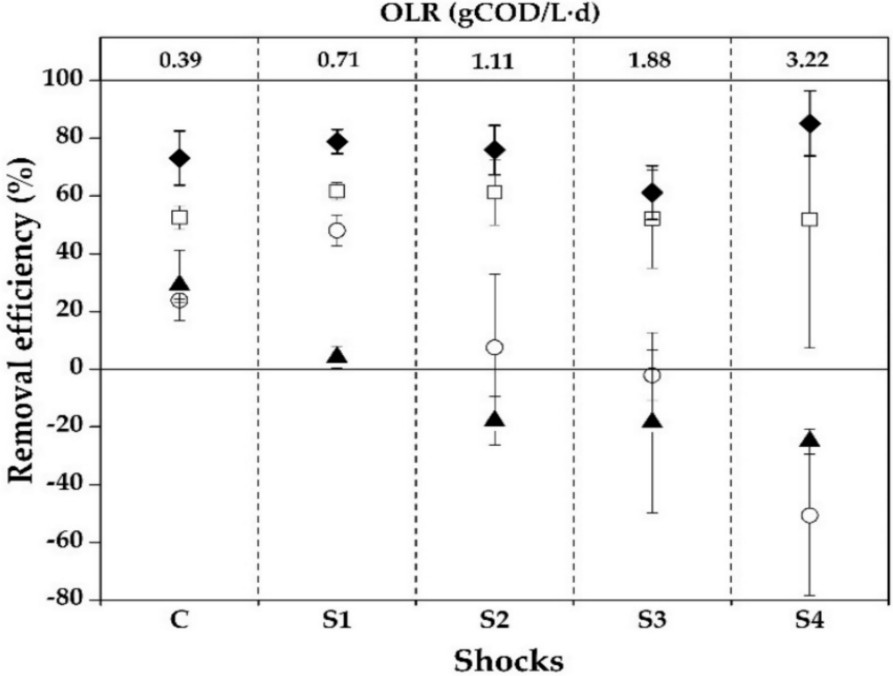

**Figure 1.** Removal efficiency for $BOD_5$ (♦), COD (□), total phenolic compounds (▲) and color (○). C: normal operating conditions, S1: shock 1, S2: shock 2, S3: shock 3, S4: shock 4.

### 3.4. Biomass and Heterotrophic Activity

Biomass was maintained at around 3.4 gVSS/L in normal system operation. VSS in S1 decreased to 3.17 gVSS/L. In S2, it decreased again to a value of 2.64 gVSS/L at 24 h, while at 48 h, it increased to 3.98 gVSS/L. S3 recovered in normal conditions, with a value of 3.45 gVSS/L. S4 presented an increase to 4.67 gVSS/L at 24 h, while at 48 h, it decreased to 3.16 gVSS/L (Figure 2). This concentration is similar to that reported by Xavier et al. [31] in a system activated with 2 to 6 gVSS/L. Biomass concentration and heterotrophic activity has an important effect on treatment performance [33]. Conventionally activated sludge systems operate at high suspended biomass concentrations. In the process of conventionally activated sludge, a sludge concentration in the mixed liquor (MLSS) within the range of 3 to 5 g/L is generated [32] Studies have shown that the minimum MLSS in the activated sludge process treatment is 2000 to 2500 mg/L [34]. The biomass heterotrophic activity is

measured through the oxygen uptake rate. Biomass activity can be affected by substrate concentration and the presence of inhibitory or toxic substances [35]. Figure 2 shows the oxygen consumption with the addition of black liquor shocks S1–S4. In all cases, a decrease in heterotrophic activity is observed, showing a momentary destabilization in the activated sludge system. Twenty four hours after shock S1, the oxygen consumption was stabilized, likely by the soundness of the system. Nevertheless, the response to S2 and S3 was an abrupt decrease, with 0.301 and 0.202 mgO$_2$/gVSS·min detected, respectively, showing a destabilizing effect of toxicity to the shock of black liquor in the system. However, the system presented evidence of resilience to increased black liquor concentrations after shock S4. A key factor was likely pH adjustment (neutrality). In this regard, Sandberg and Holby [36] observed that pH (7) bacteria could resist high black liquor concentrations before being inhibited. Vadodaria [30] confirmed that there are microorganisms that can survive toxicity shock or adapt to black liquor. Biomass concentration and activity are the main factors that determine the purification efficiency of biological wastewater treatment systems.

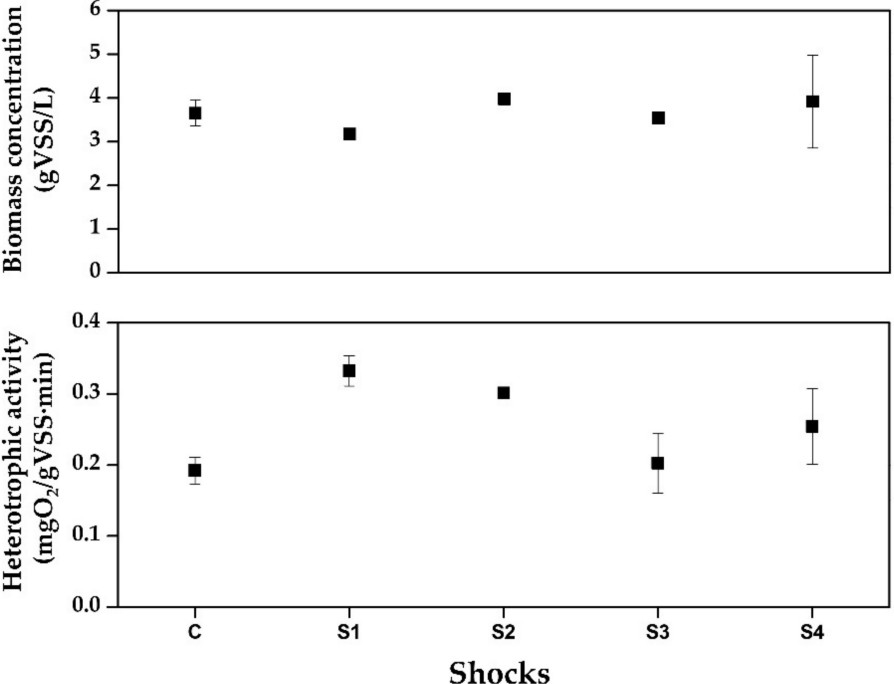

**Figure 2.** Heterotrophic activity and biomass concentration exposed to black liquor shocks. C: Normal operating conditions: S1: shock 1, S2: shock 2, S3: shock 3, S4: shock 4.

### 3.5. Toxicity Testing in Daphnia magna and Statistical Analysis

*Daphnia magna* exposed to black liquor shocks did not reveal a lethal concentration (LC$_{50}$), measured in an acute toxicity test (for 24–48 h) (data not shown). This result corroborates that AS biological treatment is capable of eliminating acute toxicity contained in the fed influent [5]. Although LC$_{50}$ is not detected, there is evidence of mortality in S3 > S2 > S1 with four, four, and three organisms in each assay, respectively, which is evident since the concentration of toxicity progressively increases. However, in S4 (30 mL black liquor/L), no mortality is observed. Thus, it is possible that the compounds contained in the black liquor have an antagonistic effect. Studies in this area suggest that cellulose effluent presents a wide range of specific compounds, including triterpenes, ketones, and phytosterols, that have a beneficial effect on organisms by decreasing acute toxicity [28]. This effect was discussed by several authors, who reported that phytosterols that are able to enter the metabolic pathway act as a natural antagonistic hormone [37–41]. In this study after 48 h of exposure *Daphnia magna* was observed to present cyclomorphosis, which is a characteristic phenomenon of seasonal or otherwise planktonic populations that consists of changes in

the size or shape of certain structures or body parts [42]. These processes usually occur due to some external environmental factor, producing morphological changes in the size of the caudal spine or helmet. In this study, *Daphnia magna* presented caudal spine and rostrum alterations as a result of sublethal toxicity (Figure 3 and Table 3). So, 30% of exposed organisms presented caudal spine and rostrum alterations when they were exposed to 10 mL and 30 mL black liquor/L, corresponding to S3 and S4 (Figure 3), in concentrations of 50 and 100%, respectively. In S3, at 100%, *Daphnia magna* presented alterations in the caudal spine (Figure 3b,c). In S4, they presented alterations in all concentrations assayed (6.25–100%) (Figure 3e). This phenomenon may be due to a response to food limitation and toxicity compounds [43], whereby upon exposure to environmental stressors, daphnids exhibit a significant reproductive decline, aberrant behavioral patterns, and ultimately, phenoplasticity [44]. Other authors explain that the corporal alteration (normally of the shell) is the most typical deformity in *Daphnia magna* associated with morphological defense [17,19,45].

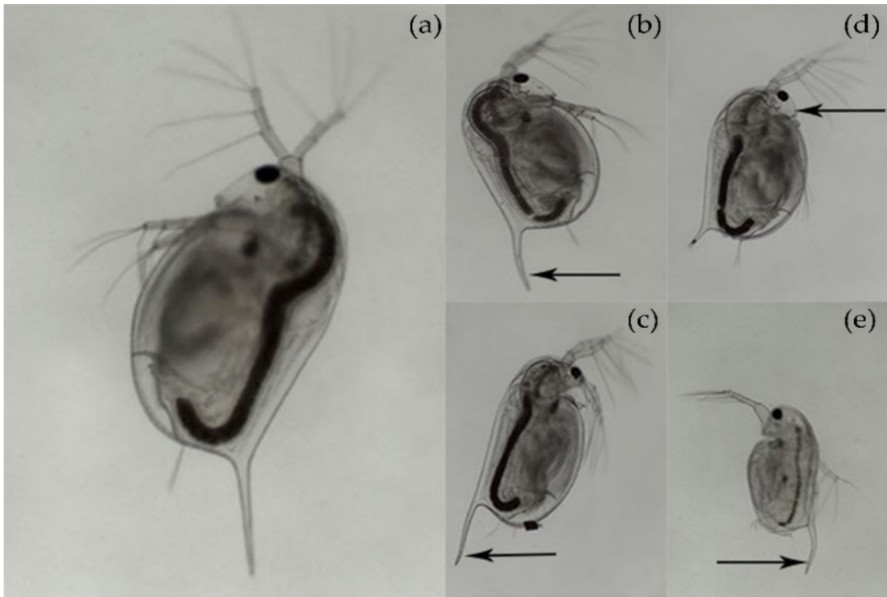

**Figure 3.** Pictures of *Daphnia magna* exposed to different concentrations of black liquor shock at bioassays end point. (**a**) Normal operating conditions; (**b**) shock S1, the arrow indicates alteration in caudal spine; (**c**) shock S2, the arrow indicates alteration in rostrum; (**d**) shock S3, the arrow indicates alteration in caudal spine; (**e**) shock S4, the arrow indicates alteration in caudal spine.

**Table 3.** Statistical analysis of the variation in *Daphnia magna* body and abdomen size patterns after exposure to kraft black liquor shock.

| Variable | $R^2$ | | Intercept | L | C | Interaction |
|---|---|---|---|---|---|---|
| Body | 0.662 | Coefficient | $5.09 \times 10^{-2}$ | $6.71 \times 10^{-5}$ | $1.24 \times 10^{-4}$ | $4.42 \times 10^{-6}$ |
| | | Standard Error | $1.08 \times 10^{-3}$ | $7.16 \times 10^{-5}$ | $2.21 \times 10^{-5}$ | $1.42 \times 10^{-6}$ |
| | | Probability | $5.08 \times 10^{-60}$ | $3.52 \times 10^{-1}$ | $2.93 \times 10^{-7}$ | $2.58 \times 10^{-3}$ |
| Abdomen | 0.645 | Coefficient | $3.60 \times 10^{-2}$ | $4.35 \times 10^{-5}$ | $7.33 \times 10^{-5}$ | $3.06 \times 10^{-6}$ |
| | | Standard Error | $7.15 \times 10^{-4}$ | $4.72 \times 10^{-5}$ | $1.45 \times 10^{-5}$ | $9.37 \times 10^{-7}$ |
| | | Probability | $2.12 \times 10^{-62}$ | $3.60 \times 10^{-1}$ | $2.82 \times 10^{-6}$ | $1.59 \times 10^{-3}$ |

L: Black liquor concentrations S1–S4 (mL black liquor/L influent); C: Bioassay concentrations (6.25–100%).

As shown in Table 3, the statistical analysis revealed that the black liquor shock trials and bioassay dilution had a significant effect on body ($R^2 = 0.64$) and abdomen size ($R^2 = 0.65$). The body/abdomen size alterations reveal that the kraft mill effluent contains compounds that have the capacity to act as hormonal moderators, principally influencing

growth [46]. In this sense, Xavier et al. [47] observed increases in the body and abdominal length of 20% and 2%, respectively, under 10% and 20% concentrations of kraft pulp mill effluent in a chronic toxicity test (21 days). However, at a concentration of 5% the length of the body and the abdomen cavity decreased by about 1% relative to the control. Moreover, López et al. [18] found deviations of 25.6–27.8% from the natural shape of *Daphnia magna* over the normal association when exposed to kraft mill effluents, and determined that phytosterols (β-sitosterol and stigmasterol) per se were responsible for 12.9% and 8.1% of shape deviation. Likewise, abdominal growth promotion is a well-known adaptive strategy that allows *Daphnia magna* to regulate its growth to maximize its fitness under variable food conditions. In general, kraft pulp mill effluents present no acute toxicity [5]; however, sublethal and chronic tests have shown there are still refractory compounds capable of interfering with the metabolism of *Daphnia* sp. [48,49].

### 3.6. Mutagenicity and Cytotoxicity Test

Black liquor mutagenicity and cytotoxicity results are shown in Table 4. S1 and S2 result in mutagenicity activity in all tested samples, with maximums of 37 and 45 revertants per plate in contrast with the control, while for the 48 h, they show cytotoxicity. For S3 and S4, cytotoxicity is observed at 24 h and 48 h; For 24 h, the S factor values for S3 and S4 are 0.375 and 0.111, respectively. This confirms that the black liquor contains extensive compounds capable of inducing mutagenicity and genotoxicity. Black liquor has a complex matrix that contains terpenes, ketones, and macromolecules such as lignin that induce toxicity (mutagenicity and genotoxicity). Savant et al. [25] warned that compounds with a low molecular weight, smaller than 1000 g/mol, pose the greatest toxicological, mutagenic, and carcinogenic risk, as they easily penetrate the cell membrane [50]. Meanwhile, Rao et al. [50] determined mutagenicity in a filtered effluent (4 mL) and found genotoxic potential at about 0.28 mL of the effluent. According to this author, most of the mutagenic were derived from polar organic compounds [51]. Moreover, specifically β-sitosterol and abietic acid have a clear genotoxic effect at concentrations higher than 8 to 15 mg/L [52,53].

**Table 4.** Mutagenicity and cytotoxicity induced by *Salmonella typhimurium* TA 100 exposed to black liquor shocks, as determined through an Ames test.

| Treatments | Revertant per Plate (+S9) | | | | | S Factor | Result |
|---|---|---|---|---|---|---|---|
| | Day | | | | | | |
| | 2 | 3 | 4 | 5 | 6 | | |
| −Control | 0 | 0 | 0 | 0 | 0 | - | - |
| Background | 0 | 13 | 16 | 18 | 20 | - | - |
| +Control | 0 | 44 | 63 | 70 | 72 | - | - |
| S1 24 h | 0 | 21 | 25 | 31 | 37 | 0.514 | Mutagenicity, 99.0% significance |
| S1 48 h | 0 | 16 | 18 | 19 | 21 | 0.292 | Possible cytotoxicity |
| S2 24 h | 0 | 25 | 43 | 43 | 45 | 0.625 | Mutagenicity, 99.9% significance |
| S2 48 h | 0 | 5 | 5 | 7 | 10 | 0.139 | Cytotoxicity |
| S3 24 h | 0 | 4 | 23 | 25 | 27 | 0.375 | Possible cytotoxicity |
| S3 48 h | 0 | 9 | 15 | 19 | 25 | 0.347 | Possible cytotoxicity |
| S4 24 h | 7 | 8 | 8 | 8 | 8 | 0.111 | Cytotoxicity |
| S4 48 h | 7 | 7 | 8 | 10 | 10 | 0.139 | Cytotoxicity |

The test was performed with the TA 100 strain, which is used to detect point mutations, and with the presence of the activation enzyme S9, which biotransforms promutagen compounds into mutagenic compounds.

Future studies should be carried out to determine the physical, chemical and toxicological characteristics of the compounds contained in the black liquor that generate mutagenic and genotoxic potential.

## 4. Conclusions

AS-fed black liquor was capable of a COD removal of 64.84%, which fell to 61.68% for S1 and 61.31 and 52.11% for S3 and S4, respectively. The minimum efficiency was observed for S4, which had a value of 20.34%. $BOD_5$ removal efficiency was up to 75% in all the shocks.

*Daphnia magna* exposed to black liquor shocks did not reveal lethal concentration media ($LC_{50}$) on samples studies; however, it presented caudal spine and rostrum deformities when it was exposed to S3 and S4. Mutagenicity was observed at 2 mL and 4 mL black liquor/L influent concentration and cytotoxicity effects at 10 mL and 30 mL black liquor/L influent.

Therefore, it is feasible to apply doses of 2 (S1) or 4 (S2) mL black liquor/L influents to an AS system that treats kraft pulp mill effluent, as S1 and S2 did not substantially affect the performance of COD removal. Moreover, at these dose levels, the generated effluent does not present any toxicity to *Daphnia magna*. Thus, it is possible to discharge the black liquor accidentally generated during the process in pulses ("doses" in this paper) of black liquor, together with the rest of the kraft mill effluents treated by an AS system. Moreover, this study also shows through the evaluation of *Daphnia magna* toxicity that doses S1 and S2 do not present toxicity; this means that there is no indication of the impact on the environment, which is also corroborated for cytotoxicity. However, mutagenicity was observed for doses S1 and S2.

This study is novel because it shows the feasibility of solving a real problem in the kraft pulp mill, with the installed infrastructure adjusting an optimal process for dosing the black liquor spill to the activated sludge system. Future studies should be carried out to determine the physical, chemical and toxicological characteristics of the specific compounds contained in the black liquor that generate mutagenic and genotoxic potential in order to more accurately describe the environmental risks of carrying out these processes.

**Author Contributions:** Conceptualization, G.V. and S.C.; methodology, S.C. and L.H.; software, G.G.; validation, G.V., S.C. and K.S.; formal analysis, K.S.; investigation, L.H. and S.C.; resources, G.V.; data curation, G.G. and S.C.; writing—original draft preparation, S.C. and G.G.; writing—review and editing, G.V., S.C. and G.G.; visualization, G.G.; supervision, G.V.; project administration, G.V.; funding acquisition, G.V. All authors have read and agreed to the published version of the manuscript.

**Funding:** This research was funded by ANID/FONDAP/15130015.

**Conflicts of Interest:** The authors declare no conflict of interest.

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
