# Peer review of "Effects of Black Liquor Shocks on the Stability of Activated Sludge Treatment of Kraft Pulp Mill Effluent: Morphological Alteration in Daphnia magna and Mutagenicity and Genotoxicity Response in Salmonella typhimurium"

_sustainability, doi:10.3390/su14073869_

Round 1

Reviewer 1 Report

Before consideration for publishing this manuscript, some critical problems should be corrected.

The English in the manuscript should be improved, the manuscript requires extensive English editing. There are numerus grammatically errors.

Most of the references are old. From 52 references, only 5 of them are dating from the last 6 years. Hod do authors explain this? Aren’t there any novelty in this research, is the presented research out of date? What are the recent findings?

In the introduction part of the manuscript, the authors started with the information related to treatment technologies, but from my point of view, first the information regarding the problems should be stated, afterwards, the treatment technologies should be named.

Line 35: environmental legislation and standards, not norms

Lines 35-37: The conventional treatment technologies that are currently installed in all Kraft pulp mills are of the conventional, biological type and all discharge their effluents to surface water bodies [1] – this statement should be checked. Is this information reliable? Are all mills equipped with those technologies… are all effluents discharged to water bodies? What about the public sewerage system?

Line 39: …to remove aerobically organic matter from effluents… - this means that organic matter is aerobic… please correct this sentence within the “spirit” of English language. For example, “…for the aerobic degradation of organic compounds present in effluents.

Line 40-41 – COD can’t be removed, it can be reduced. It is a parameter. In addition, you have found only one reference dealing with the reduction of COD?

Line 42: have been identified (numerous errors like this are present in the manuscript, it should be corrected)

Line 43: toxic nature derives (numerous errors like this are present in the manuscript, it should be corrected)

Lines 45-47: Specifically black liquor generated in the digestion stage and characterized by inorganic cooking elements and the degraded dissolved wood substance [7, 8] -  a sentence like this has no meaning, I thing the grammar is problematic. Should be checked.

Line 52: shock or inhibit? Is a shock a proper word? From my point of view, it is a slang

Line 63: Daphnia should be written in italic , and capital letter

Line 99: why was the presented proportion of BOD:N:P chosen? There is no explanation

Line 101: HCl (0.5 N) or 0.5 N HCl.

Lines 102-103: how was the concentration of DO monitored, if you say it was below 2 mg/L?

Line 109: denominated as

Lines 109 – 110: the concentrations of black liquor added were 2, 4, 10 and 30 mL of black liquor / L of influent, respectively  - this is not correct format of concentration. the presentation of concentration should be corrected. If you have 30 mL of BL in 1L of influent, than what concentration do you have?

Line 113: HRT meaning, explain the abbreviation when first time used

Line 130: despite the reference given, some information regarding the used method for determination of COD and BOD should be given

Lines 131- 137: why were the presented wavelengths used for the determination of mentioned parameters? Why for example the 440 nm was used for colour measurement?  Why haven’t you conducted some relevant analytical procedure? This refers to my comment on results obtained. As mentioned in the lines 188-197: what meaning od colour measurement 0.1 +-0.03 1x1 cm? First time I see those results and I am not source what do they present. The colour of the samples should be measured through the whole VIS spectral range and get the spectral reflection curve form which the LAB colour parameters should be calculated. Moreover, Colour difference can be expressed from those measurements. Additionally, the calculation of lignin, aromatic compounds, lignosulfonic acid- why haven’t you provide the concentrations. Result presented in this form are not relevant and can’t be used for the physicochemical characterization.

Line 165: Daphnia magna were observed (48 h) via photographic under microscope 40x. - Daphnia magna was observed (after 48 h or what?) with microscope under magnification of 40x and microscopic images were obtained. – missing the information about microscope, optical or some else? And microscope producer as well.

Lines 213-214: Table 2: Missing the BOD values her hence you have later the description of BOD efficiency removal. In addition, why haven‘t you provided all the measurements presented in table 1 – lignin, aromatic compounds, etc?

Line 212 in table 2 dosage of Black liquor should be mL/L not ml/L

Lines 208-212: how the mentioned statement has the connection with your research, why it is here?

Lines 222-227- how BOD removal was conducted since you haven’t presented those measurements?

Lines 232-233: to the presence of lignins of high molecular weight and the presence of phenolic chlorate compounds that are difficult to degrade – this statement is not supported by the measurable concentrations

Lines 233-237: this statement is not supported by the measurable concentrations

Lines 238-241: the measurements of BOD haven’t been shown in the manuscript

Line 315: missing the scale bare in the presented microscopic image

Lines 348-356:The conclusion is minor and should be in more details. There are no important details in it.

Author Response

Mr. Filip Petrović,

Assistant Editor

Sustainability Journal

Concepción (Chile), February 17th, 2022

Dear Editor,

Please find here with the revised version of the manuscript “Effects of black liquor shocks on stability of activated sludge treatment of kraft pulp mill effluent: Morphological alteration in Daphnia magna and mutagenicity and genotoxicity response in Salmonella typhimurium” by Soledad Chamorro, Laura Hernández, Katia Saéz, Gloria Gómez and Gladys Vidal Manuscript ID: sustainability-1579440.

We want to thank the reviewers for carefully reading the manuscript. All the comments were attended and the paper was carefully checked according to the editorial suggestions.

The following comments indicate our response to all of the questions and notes indicated by the Reviewer #1. Please consider that unless otherwise expressed, the referenced lines correspond to the current numeration after the corrections were made.

REVIEWER # 1

- Comment 1: The English in the manuscript should be improved, the manuscript requires extensive English editing. There are numerus grammatically errors.

- Answer: The English was improved for all the manuscript with a native speaker.

- Comment 2: Most of the references are old. From 52 references, only 5 of them are dating from the last 6 years. Hod do authors explain this? Aren’t there any novelty in this research, is the presented research out of date? What are the recent findings?

- Answer: An update of the publications was made, as far as posible. An update of the publications was made, as far as possible. Updated paper were highlighted in yellow

- Comment 3: In the introduction part of the manuscript, the authors started with the information related to treatment technologies, but from my point of view, first the information regarding the problems should be stated, afterwards, the treatment technologies should be named.

- Answer: The introduction was improved.

- Comment 4: Line 35: environmental legislation and standards, not norms

- Answer: Thank you very much. The change was made.

- Comment 5: Lines 35-37: The conventional treatment technologies that are currently installed in all Kraft pulp mills are of the conventional, biological type and all discharge their effluents to surface water bodies [1] – this statement should be checked. Is this information reliable? Are all mills equipped with those technologies… are all effluents discharged to water bodies? What about the public sewerage system?

- Answer: The Kraft pulp mill effluent due to the level of flow (thousands of liters / second) does not discharge into sewage systems, they have their own treatment systems and surface discharges into the river and/or sea. In fact, Kraft pulp mills are working in open circuit and have aerobic biological type primary and secondary treatment systems.

- Comment 5: Line 39: …to remove aerobically organic matter from effluents… - this means that organic matter is aerobic… please correct this sentence within the “spirit” of English language. For example, “…for the aerobic degradation of organic compounds present in effluents.

- Answer: The sentence was improved.

- Comment 6: Line 40-41 – COD can’t be removed, it can be reduced. It is a parameter. In addition, you have found only one reference dealing with the reduction of COD?

- Answer: A reference was added

- Comment 7: Line 42: have been identified (numerous errors like this are present in the manuscript, it should be corrected) and Line 43: toxic nature derives (numerous errors like this are present in the manuscript, it should be corrected)

- Answer:

- Comment 8: Lines 45-47: Specifically black liquor generated in the digestion stage and characterized by inorganic cooking elements and the degraded dissolved wood substance [7, 8] - a sentence like this has no meaning, I thing the grammar is problematic. Should be checked.

- Answer: The sentence was modified

- Comment 9: Line 52: shock or inhibit? Is a shock a proper word? From my point of view, it is a slang

- Answer:

- Comment 10: Line 63: Daphnia should be written in italic, and capital letter

- Answer: Daphnia sp. is written italic way.

- Comment 11: Line 99: why was the presented proportion of BOD:N:P chosen? There is no explanation

- Answer: The standard operation for activated sludge is BOD:N:P

- Comment 12: Line 101: HCl (0.5 N) or 0.5 N HCl.

- Answer: The setence was changed to “HCl (0.5 N)”

- Comment 13: Lines 102-103: how was the concentration of DO monitored, if you say it was below 2 mg/L?

- Answer: The setence was changed to “The dissolved oxygen (DO) concentration was maintained at values higher than 2 mg/L, using a diffuser aeration system [2].”

-Comment 14: Lines 109 – 110: the concentrations of black liquor added were 2, 4, 10 and 30 mL of black liquor / L of influent, respectively  - this is not correct format of concentration. the presentation of concentration should be corrected. If you have 30 mL of BL in 1L of influent, than what concentration do you have?

- Answer: The notation used is maintained to compare with other publications that apply shock of toxic substances to treatment systems and evaluate toxicity through bioindicators.

- Comment 15: Line 113: HRT meaning, explain the abbreviation when first time used

- Answer: Hydraulic retention time for HRT.

- Comment 16: Line 130: despite the reference given, some information regarding the used method for determination of COD and BOD should be given

- Answer: Information was added.

- Comment 17: Lines 131- 137: why were the presented wavelengths used for the determination of mentioned parameters? Why for example the 440 nm was used for colour measurement?  Why haven’t you conducted some relevant analytical procedure? This refers to my comment on results obtained. As mentioned in the lines 188-197: what meaning od colour measurement 0.1 +-0.03 1x1 cm? First time I see those results and I am not source what do they present. The colour of the samples should be measured through the whole VIS spectral range and get the spectral reflection curve form which the LAB colour parameters should be calculated. Moreover, Colour difference can be expressed from those measurements. Additionally, the calculation of lignin, aromatic compounds, lignosulfonic acid- why haven’t you provide the concentrations. Result presented in this form are not relevant and can’t be used for the physicochemical characterization.

- Answer: Thank very much for the information. Color measurement is performed according to standards methods. Also, all researchers working in the area of Kraft pulp mill effluents use this methodology. It is not possible to give color concentrations of a specific compound, since the color is made up of different aromatic structures coming from the oxidation of lignin.

The notation in the lines 188-197 of the color measurement 1x1 cm, corresponds to the value followed by the dimension of the cell with which it was measured; in the text” 1x1 cm quartz cell using a model Spectronic Unicam UV–Visible Series GenesysTM 10 according to Chamorro et al. [26]”. Also more explanation in Kortekass et al. 2005 and the other references.

- Kortekaas, S., Vidal, G., Yan-Ling, H., Lettinga, G. and Field, J. 1998. Anaerobic-aerobic treatment of toxic pulping black liquor with upfront effluent recirculation. Journal of Fermentation and Bioengineering 86(1), 97-110.

- Chamorro S., Xavier C. and Vidal G. 2005. Behaviour of aromatic compounds contained in the kraft mill effluents measurements by UV-VIS. Biotechnology Progress 21(5), 1567-571.

- Milestone, C.B.; Fulthorpe, R.R.; Stuthridge, T.R.2004. The formation of colour during biological treatment of pulp and paper wastewater. Water Sci. Technol.  50, 87–94.

- Villamar, C.A., Jarpa, M., Decap J. and Vidal. G. 2009. Aerobic moving bed bioreactor treating kraft mill effluents from Pinus radiata and Eucalyptus globulus as raw material. Water Science and Technology 59 (3), 507–514

- Comment 18: Line 165: Daphnia magna were observed (48 h) via photographic under microscope 40x. - Daphnia magna was observed (after 48 h or what?) with microscope under magnification of 40x and microscopic images were obtained. – missing the information about microscope, optical or some else? And microscope producer as well.

- Answer: The adults Daphnia magna have a range from less than 1 mm to 5 mm in size, due to this, their measurement is by means of magnifying glass. All tests were performed according to the procedure in accordance with:

- USEPA. Short-term methods for estimating the chronic toxicity of effluents and receiving water to freshwater organisms. EPA-600-4-91-002, Washington, 1994.

- Comment 19: Lines 213-214: Table 2: Missing the BOD values her hence you have later the description of BOD efficiency removal. In addition, why haven‘t you provided all the measurements presented in table 1 – lignin, aromatic compounds, etc?

- Answer: Table 2 only provides COD values, because it is interesting to show an activated sludge treatment plant operator how much the OLR value is. The control parameter of an activated sludge plant is the OLR. For the control in the plant of accidental black liquor discharges, it is only necessary to control the COD. The BOD5 measurement is not used in the plant because the response of this parameter is very slow (exactly 5 days)

- Comment 20: Line 212 in table 2 dosage of Black liquor should be mL/L not ml/L

- Answer: The correction was made.

- Comment 21: Lines 208-212: how the mentioned statement has the connection with your research, why it is here?

- Answer: The statement was improved.

- Comment 25: Lines 222-227- how BOD removal was conducted since you haven’t presented those measurements?

- Answer: The BOD was measured at the influent and effluent of the process with this it is possible to obtain the elimination of the BOD removal.

- Comment 26: Lines 232-233: to the presence of lignins of high molecular weight and the presence of phenolic chlorate compounds that are difficult to degrade – this statement is not supported by the measurable concentrations

- Answer: You are right, but this phrase is supported by other studies that the group has been carrying out since 2000 to date in this area. Plase see references [1], [5] and [8] amoung the others. This information was add in the phrase.

- Comment 27: Lines 233-237: this statement is not supported by the measurable concentrations

- Answer: The statement was improved.

- Comment 28: Lines 238-241: the measurements of BOD haven’t been shown in the manuscript

- Answer: DOB5 is given in Table 1 and Figure 1. In the case of Figure 1, the vale of DOB5 is give has a percenatge of removal.

- Comment 29: Lines 348-356: The conclusion is minor and should be in more details. There are no important details in it.

- Answer: The conclusion was rewritten

Yours sincerely,

Reviewer 2 Report

This study evaluates the stability of activated sludge in the treatment of Kraft pulp mill effluent. Please consider the following comments to improve the manuscript.

Comments:

  1. The first and foremost issue is language of the manuscript. There are several grammatical and punctuation related errors throughout the manuscript. Also, authors have framed very long sentences throughout the manuscript which makes the sentences confusing and sometimes completely vague. So authors must check the manuscript language and also rephrase the long sentences throughout the manuscript.
  2. An abstract should have the following in this order: Introductory/rationale statement that leads to an objective statement (should start like this "Therefore, the objective (s) was...."), then a description of the experimental design/treatments measurements and ending with the largest amount of the abstract text as results/major conclusions. The description of the experimental design/treatments measurements is Ok, but the authors need to improve the other points indicated. In the conclusion section, authors should emphasize the novelty of the research rather than summarizing findings. Please rewrite it.
  3. Provide significant words which are more relevant to the work in logical sequence as ‘keywords’. Also use keywords which are not present in title.
  4. What is the current level of understanding in relation to the mutagenicity and genotoxicity of Kraft pulp mill effluent? What are the knowledge gaps?. These should be included in the introduction section. The introduction is insufficient to provide the state of the art in the topic. Hypothesis should be given. How this work is different from the available data?
  5. The introduction of the paper must be extended and reformulated in order to provide a more comprehensive approach. Add recent references accordingly.
  6. It would be necessary to develop more bioinformatic/statistical analyses in the present study. 
  7. The discussion and interpretation of results does not clearly explain its impact on the literature and the field. Authors are suggested to add discussion by explaining trends in the obtained results along with the possible mechanisms behind the trends.
  8. Add a paragraph 'practical implications of this study,' outlining the challenges in the current research, future work, and recommendations, before the conclusion.
  9. Conclusion: Conclusions is not just about summarizing the key results of the study, it should highlight the insights and the applicability of your findings/results for further work.  Please enrich your conclusion.

Author Response

Dear Editor,

Please find here with the revised version of the manuscript “Effects of black liquor shocks on stability of activated sludge treatment of kraft pulp mill effluent: Morphological alteration in Daphnia magna and mutagenicity and genotoxicity response in Salmonella typhimurium” by Soledad Chamorro, Laura Hernández, Katia Saéz, Gloria Gómez and Gladys Vidal Manuscript ID: sustainability-1579440.

We want to thank the reviewers for carefully reading the manuscript. All the comments were attended and the paper was carefully checked according to the editorial suggestions.

The following comments indicate our response to all of the questions and notes indicated by the Reviewer #2. Please consider that unless otherwise expressed, the referenced lines correspond to the current numeration after the corrections were made.

REVIEWER # 2

- Comment 1. The first and foremost issue is language of the manuscript. There are several grammatical and punctuation related errors throughout the manuscript. Also, authors have framed very long sentences throughout the manuscript which makes the sentences confusing and sometimes completely vague. So authors must check the manuscript language and also rephrase the long sentences throughout the manuscript.

- Answer: The manuscript was cheked by a native speaker.

- Comment 2. An abstract should have the following in this order: Introductory/rationale statement that leads to an objective statement (should start like this "Therefore, the objective (s) was...."), then a description of the experimental design/treatments measurements and ending with the largest amount of the abstract text as results/major conclusions. The description of the experimental design/treatments measurements is Ok, but the authors need to improve the other points indicated. In the conclusion section, authors should emphasize the novelty of the research rather than summarizing findings. Please rewrite it.

- Answer: The abstract was rewritten.

- Comment 3. Provide significant words which are more relevant to the work in logical sequence as ‘keywords’. Also use keywords which are not present in title.

- Answer: Keywords were changed by: Toxicity of black liquor; performance of activated sludge; organic matter biodegradation; heterotrophic biomass activity, effluent toxicity.

- Comment 4. What is the current level of understanding in relation to the mutagenicity and genotoxicity of Kraft pulp mill effluent? What are the knowledge gaps?. These should be included in the introduction section. The introduction is insufficient to provide the state of the art in the topic. Hypothesis should be given. How this work is different from the available data?

- Answer: In this paper the authors intend to evaluate the mutagenicity and genotoxicity of the effluents generated when black liquor is applied to an activated sludge system. However, this study does not intend to provide a status of the progress of these issues.

This work aims to provide a solution to a problem commonly encountered in the operation of Kraft pulp effluent treatment plants.

Despite this, the authors make a review of the works that show mutagenicity / genotoxicity as described in the paragraphs:

"Several types of mutagenic compounds have been identified in the pulp kraft mill effluent [20]. High molecular weight compounds generate organic matter that is difficult to biodegrade in biological systems; therefore, color is present in the final effluent [1, 21]. cell membrane.Therefore, by damaging the DNA of organisms, genotoxins can initiate a cascade of impairments at the molecular, cellular, organ, whole organism, or population and community levels.DNA and cyto-genetic alterations in aquatic organisms have been associated with an impaired en-zyme function or general metabolism, cytotoxicity, immunotoxicity, abnormal devel-opment, and reduced survival, growth, and reproductive potency ………………….”.

- Comment 5. The introduction of the paper must be extended and reformulated in order to provide a more comprehensive approach. Add recent references accordingly.

- Answer: Recent references were add. It is about adding the newest bibliography on the subject, it is what is in the specific field of “Kraft pulp mil”.

- Comment 6. It would be necessary to develop more bioinformatic/statistical analyses in the present study. 

- Answer: I apologize, but with the data we have we cannot do deeper bioinformatics/statisctical studies. The data of this work were carefully worked, please see the section "Statistical analysis"

- Comment 7. The discussion and interpretation of results does not clearly explain its impact on the literature and the field. Authors are suggested to add discussion by explaining trends in the obtained results along with the possible mechanisms behind the trends.

- Answer: The discussion was improved.

- Comment 8. Add a paragraph 'practical implications of this study,' outlining the challenges in the current research, future work, and recommendations, before the conclusion.

- Answer: The paragraph 'practical implications of this study” was add in the abstract and conclusions and also after objectives.

- Comment 9. Conclusion: Conclusions is not just about summarizing the key results of the study, it should highlight the insights and the applicability of your findings/results for further work. Please enrich your conclusion.

- Answer: Conclusion was improved.

Yours sincerely,

Reviewer 3 Report

Lines 51-53 (In these sense, spills of black liquor on receiving waters, are sources of air emission, and can shock the microbial action of wastewater treatment system): The sentence is unclear, and the grammar needs to be improved.

Lines 56-59 (The assessment of biological effects of wastewater discharges in the ecosystems is consider as relevant evidence to identify the ecological hazard of environmental impacts moreover, there is relatively little published information on the unplanned discharges): Please, improve the grammar. Furthermore, could you present a list of case studies where such wastewater was discharged in the ecosystems causing environmental and health issues?

Lines 84-87 (The objective of this study is to evaluate the stability of activated sludge in the treatment of Kraft pulp mill effluent, exposed to black liquor shock, as well as the effect of their exposure on the morphology of Daphnia magna and the DNA damage through mutagenicity and genotoxicity response in Salmonella typhimurium): At the end of the Introduction section, you wrote the objective of your study. However, you should highlight the novelty (if any) compared to previous research and the consequent contribution of your work to the scientific community.

Section 4 – Conclusions: The section is very short (only seven lines!) and inconclusive. Perspectives from this study are not mentioned. Furthermore, novelty (if any) and contribution with respect to previous research were not discussed. If the authors do not want to improve the conclusions, they should at least add such information in the Discussion section. Indeed, in the “instructions for authors” they can read (concerning the Discussion section): “Authors should discuss the results and how they can be interpreted in perspective of previous studies and of the working hypotheses. The findings and their implications should be discussed in the broadest context possible and limitations of the work highlighted. Future research directions may also be mentioned.”

Author Response

Concepción (Chile), February 17th, 2022

Dear Editor,

Please find here with the revised version of the manuscript “Effects of black liquor shocks on stability of activated sludge treatment of kraft pulp mill effluent: Morphological alteration in Daphnia magna and mutagenicity and genotoxicity response in Salmonella typhimurium” by Soledad Chamorro, Laura Hernández, Katia Saéz, Gloria Gómez and Gladys Vidal Manuscript ID: sustainability-1579440.

We want to thank the reviewers for carefully reading the manuscript. All the comments were attended and the paper was carefully checked according to the editorial suggestions.

The following comments indicate our response to all of the questions and notes indicated by the Reviewer # 3. Please consider that unless otherwise expressed, the referenced lines correspond to the current numeration after the corrections were made.

REVIEWER # 3

- Comment 1: Lines 51-53 (In these sense, spills of black liquor on receiving waters, are sources of air emission, and can shock the microbial action of wastewater treatment system): The sentence is unclear, and the grammar needs to be improved.

- Answer: The gramar was improved: (However, after the accident, the only way to discharge these spills is through the effluent treatment plant [2])

- Comment 2: Lines 56-59 (The assessment of biological effects of wastewater discharges in the ecosystems is consider as relevant evidence to identify the ecological hazard of environmental impacts moreover, there is relatively little published information on the unplanned discharges): Please, improve the grammar. Furthermore, could you present a list of case studies where such wastewater was discharged in the ecosystems causing environmental and health issues?

- Answer: The English of the manuscript was improved.

- Comment 3: Lines 84-87 (The objective of this study is to evaluate the stability of activated sludge in the treatment of Kraft pulp mill effluent, exposed to black liquor shock, as well as the effect of their exposure on the morphology of Daphnia magna and the DNA damage through mutagenicity and genotoxicity response in Salmonella typhimurium): At the end of the Introduction section, you wrote the objective of your study. However, you should highlight the novelty (if any) compared to previous research and the consequent contribution of your work to the scientific community.

- Answer: Now the novel was included.

“This study is novel because a problem that arises in kraft pulp mills could be solved, with the infrastructure installed (through the installed treatment plant) adjusting an optimal process for dosing the black liquor spill to the activated sludge system. In this way, maintaining a stable operation in the activated sludge system will generate discharges suitable for surface ecosystems. In addition to evaluating the stability of the AS operation with biomass activity tools, this study also provides results on the toxicity of the treated effluent (mixture of the effluent from the plant process plus a dose of the black liquor spill) discharged into ecosystems.”

- Comment 4: Section 4 – Conclusions: The section is very short (only seven lines!) and inconclusive. Perspectives from this study are not mentioned. Furthermore, novelty (if any) and contribution with respect to previous research were not discussed. If the authors do not want to improve the conclusions, they should at least add such information in the Discussion section. Indeed, in the “instructions for authors” they can read (concerning the Discussion section): “Authors should discuss the results and how they can be interpreted in perspective of previous studies and of the working hypotheses. The findings and their implications should be discussed in the broadest context possible and limitations of the work highlighted. Future research directions may also be mentioned.”

- Answer: The section was improved.

Yours sincerely,

Round 2

Reviewer 1 Report

The authors should have made all the changes in some different colur. In this form I do not see what they have changed.

Some of the comments authors have ignored.

Like, HRT abbreviation. When you first time use some abbreviation than pleas explain it first time. You did not do thaht.

In addition you havent answered all my questions.

Author Response

Dear Reviewer,

The answers are in the file attached.

Best regards,

Gladys

Reviewer 2 Report

The authors addressed comments in the revised version, but there are still some queries which needs to be addressed.

  1. The objectives of the study should be properly discussed in the last paragraph of the introduction section.
  2. It would be better to use more recent references in discussion part. 
  3. Some related lines of future research should be incorporated after results and discussion part.

  4. Rewrite conclusion and give significance of the study. 

Author Response

The authors addressed comments in the revised version, but there are still some queries which needs to be addressed.

- Comment 1. The objectives of the study should be properly discussed in the last paragraph of the introduction section.

- Answer: The objectives of the study was discussed in the last paragraph of the introduction section and what is new in the study is highlighted in yellow.

- Comment 2. It would be better to use more recent references in discussion part. 

- Answer: We have updated references, however the topic is very specific. We have made the best effort. Updated references are highlighted in yellow.

- Comment 3. Some related lines of future research should be incorporated after results and discussion part.

- Answer: A sentence was included to indicate future perspectives of this study.

- Comment 4. Rewrite conclusion and give significance of the study. 

- Answer: The conclsuion was rewrite.

Reviewer 3 Report

Dear Authors,
You did not use the track-changes mode to highlight the changes in the manuscript. In general, you did not use any different colours.
As a consequence, it appears impossible for me to conduct an appropriate review of the manuscript. 
I kindly ask you to send a suitable manuscript for revision using the track-change mode or different colours to verify what you have changed.
Thank you.

Author Response

REVIEWER # 3

Revision 2

- Comment 1: You did not use the track-changes mode to highlight the changes in the manuscript. In general, you did not use any different colours. As a consequence, it appears impossible for me to conduct an appropriate review of the manuscript.

I kindly ask you to send a suitable manuscript for revision using the track-change mode or different colours to verify what you have changed. Thank you.

- Answer: My apologies. Now the changes were highlighted in yellow. It is important to indicate that all the text is changed, due to the correction of the English by a native sapeker

Revision 1

- Comment 1: Lines 51-53 (In these sense, spills of black liquor on receiving waters, are sources of air emission, and can shock the microbial action of wastewater treatment system): The sentence is unclear, and the grammar needs to be improved.

- Answer: The gramar was improved: (However, after the accident, the only way to discharge these spills is through the effluent treatment plant [2])

- Comment 2: Lines 56-59 (The assessment of biological effects of wastewater discharges in the ecosystems is consider as relevant evidence to identify the ecological hazard of environmental impacts moreover, there is relatively little published information on the unplanned discharges): Please, improve the grammar. Furthermore, could you present a list of case studies where such wastewater was discharged in the ecosystems causing environmental and health issues?

- Answer: The English of the manuscript was improved.

- Comment 3: Lines 84-87 (The objective of this study is to evaluate the stability of activated sludge in the treatment of Kraft pulp mill effluent, exposed to black liquor shock, as well as the effect of their exposure on the morphology of Daphnia magna and the DNA damage through mutagenicity and genotoxicity response in Salmonella typhimurium): At the end of the Introduction section, you wrote the objective of your study. However, you should highlight the novelty (if any) compared to previous research and the consequent contribution of your work to the scientific community.

- Answer: Now the novel was included.

“This study is novel because a problem that arises in kraft pulp mills could be solved, with the infrastructure installed (through the installed treatment plant) adjusting an optimal process for dosing the black liquor spill to the activated sludge system. In this way, maintaining a stable operation in the activated sludge system will generate discharges suitable for surface ecosystems. In addition to evaluating the stability of the AS operation with biomass activity tools, this study also provides results on the toxicity of the treated effluent (mixture of the effluent from the plant process plus a dose of the black liquor spill) discharged into ecosystems.”

- Comment 4: Section 4 – Conclusions: The section is very short (only seven lines!) and inconclusive. Perspectives from this study are not mentioned. Furthermore, novelty (if any) and contribution with respect to previous research were not discussed. If the authors do not want to improve the conclusions, they should at least add such information in the Discussion section. Indeed, in the “instructions for authors” they can read (concerning the Discussion section): “Authors should discuss the results and how they can be interpreted in perspective of previous studies and of the working hypotheses. The findings and their implications should be discussed in the broadest context possible and limitations of the work highlighted. Future research directions may also be mentioned.”

- Answer: The section was improved.

Round 3

Reviewer 1 Report

in Figure 3. Microscopy images of Daphnia magna exposed at different concentrations of black liquor shock are presented. But in methodology part you haven't named what kind of microscopy was used and the conditions of sampling. in lines171-172 you have: Daphnia magna were observed (48 h) via photographic under 40x microscope to observe the modification endpoints such rostrum and caudal spine. But this is not enough and changes have to be made. Additionally, the English used is wrong.

Author Response

REVIEWER # 1

- Comment 1: in Figure 3. Microscopy images of Daphnia magna exposed at different concentrations of black liquor shock are presented. But in methodology part you haven't named what kind of microscopy was used and the conditions of sampling. in lines171-172 you have: Daphnia magna were observed (48 h) via photographic under 40x microscope to observe the modification endpoints such rostrum and caudal spine. But this is not enough and changes have to be made. Additionally, the English used is wrong.

- Answer: The metodology was rewritten as follows “Daphnia magna were observed using a light microscope fitted with a photographic camera. The daphnids were placed on a glass microscope slide, immobilized by removing the medium from the slide, and anatomical development was recorded in photographs. The dimensions of abdominal cavities ……. “

Yours sincerely,

Reviewer 2 Report

The authors addressed all the comments, therefore it may be published in the current form.  

Author Response

Dear Editor,

The Reviewer 2 says OK.

With my best regards,

Gladys Vidal

Reviewer 3 Report

Dear authors,

I am glad that you improved the manuscript.

However, I still keep most of my doubts on the Conclusions. In particular:

- Comment 4: Section 4 – Conclusions: Novelty (if any) and contribution taking into account previous research were not discussed. If the authors do not want to improve the conclusions, they should at least add such information in the Discussion section.  If the authors do not want to improve the conclusions, they should at least add such information in the Discussion section. Indeed, in the “instructions for authors” they can read (concerning the Discussion section): “Authors should discuss the results and how they can be interpreted in perspective of previous studies and of the working hypotheses. The findings and their implications should be discussed in the broadest context possible and limitations of the work highlighted. Future research directions may also be mentioned.”

Author Response

REVIEWER # 3

- Comment 1: I am glad that you improved the manuscript. However, I still keep most of my doubts on the Conclusions. In particular:

- Comment 4: Section 4 – Conclusions: Novelty (if any) and contribution taking into account previous research were not discussed. If the authors do not want to improve the conclusions, they should at least add such information in the Discussion section.  If the authors do not want to improve the conclusions, they should at least add such information in the Discussion section. Indeed, in the “instructions for authors” they can read (concerning the Discussion section): “Authors should discuss the results and how they can be interpreted in perspective of previous studies and of the working hypotheses. The findings and their implications should be discussed in the broadest context possible and limitations of the work highlighted. Future research directions may also be mentioned.”

- Answer: The following paragraph was added to the conclusion:This study is novel because it shows the feasibility of solving a real problem in the Kraft pulp mill, with the installed infrastructure adjusting an optimal process for dosing the black liquor spill to the activated sludge system. Future studies should be carried out to determine the physical, chemical and toxicological characteristics of the specific compounds contained in the black liquor that generate mutagenic and genotoxic potential in order to more accurately describe the environmental risks of carrying out these processes.”

Yours sincerely,